# Effect of Magnetic Stirring on Microstructures and Properties of Ag–1.5Cu–1.0Y Alloy

**DOI:** 10.3390/ma15155237

**Published:** 2022-07-29

**Authors:** Desheng Zhang, Hongying Yang, Qin Zhang

**Affiliations:** 1School of Chemistry and Chemical Engineering, Shandong University of Technology, Zibo 255049, China; zhdsh@sdut.edu.cn; 2School of Metallurgy, Northeastern University, Shenyang 110819, China

**Keywords:** Ag–1.5Cu–1.0Y alloy, permanent magnetic stirring, rotation speed, microstructure, properties

## Abstract

The microstructure of alloys is an important factor that affects their application. In this work, the Ag–1.5Cu–1.0Y alloys were prepared by the permanent magnet stirring method at different rates. The secondary dendrite arm spacing, dendritic segregation, density, microhardness, electrical resistivity, and sulfuration corrosion resistance were analyzed to investigate the effects of different rotation speeds on the microstructures and properties of the Ag–1.5Cu–1.0Y alloy. The results showed that the primary dendrite was refined and the secondary dendrite arm spacing decreased with the increase in the stirring rate of the permanent magnets. The Ag–1.5Cu–1.0Y alloys prepared with a 900 r/min stirring rate had the largest microhardness, relatively high density, and the best sulfuration corrosion resistance. However, the stirring rate had little effect on the electrical resistivity of the Ag–1.5Cu–1.0Y alloys. To sum up, the Ag–1.5Cu–1.0Y alloy had the best comprehensive properties when the permanent magnet stirring rate was 900 r/min, including the most refined dendrites, relatively high density, the largest microhardness, and the best sulfuration corrosion resistance. The study of the effects of permanent magnet stirring speed on the microstructures and properties of the Ag–1.5Cu–1.0Y alloy provides an experimental basis for future alloy casting optimization and property improvement of silver-based alloys.

## 1. Introduction

Silver (Ag) has been widely used in jewelry materials since ancient times and up until modern society, due to its high reflectivity and attractive bright white color [1]. More notably, Ag has the highest reflectance in the visible region, thermal conductivities, and the lowest resistivity among all metals [2,3]. Therefore, it is widely used in the field of target materials. The target materials were coated on the substrate by magnetron sputtering technology, used for the production of liquid crystal displays, optical recording media, low-radiation glass electrode films or reflective layers, etc. [4,5]. For instance, Ag–Yb and Ag–Mg alloys can be fabricated into plasmonic nanostructures and transparent organic light-emitting diodes [6]. A silver–antimony–boron alloy was applied to electrical contacts and it showed high wear resistance, sulfuration corrosion resistance, and microhardness [7]. Silver–tungsten refractory materials are mainly used in industrial and domestic circuit-breaker products, due to their good weld and erosion-resistant properties [8]. Furthermore, with the rapid development of modern electronic science and technology and the increase in target materials’ market demand, the quality requirements of alloy target materials are getting higher and higher.

Ag-based alloys are the source material for magnetron sputtering. Pure Ag has the disadvantages of poor hardness, easy oxidation, and sulfuration. Copper (Cu) could increase the hardness [9,10], tensile strength [10], and oxidation resistance [11] of the alloy matrix. The addition of rare earth elements can significantly improve the mechanical properties of alloys [12,13,14]. Yttrium (Y) addition was reported to significantly enhance the toughness and isotropy, and significantly decrease the differences between the longitudinal and transverse textures of offshore engineering steel [15]. In addition, the addition of Y could enhance the oxidation resistance [16], refinement and homogenization of eutectic phases [17,18], improve tensile strength, hardness and elongation, the creep rupture life and ductility of the alloy [18], and etc.

Ag-based alloys’ properties directly affect the quality of the coating film [19]. Moreover, the grain refinement and high density of Ag-based alloys help to improve the efficiency of magnetron sputtering and the quality of alloy film. However, a series of defects, including coarse grains, segregation, porosity, and shrinkage, exist in the traditional casting process of Ag-based alloys. These defects affect the Ag-based alloys’ mechanical properties and service life, leading to a disadvantage in the subsequent processing of the material. The addition of Cu and Y may reduce a part of the above defects. Thus, the Ag–1.5Cu–1.0Y alloy was designed and used in this study.

In addition, methods including mechanical stirring [20], permanent magnet stirring [21], electromagnetic stirring [22], twin-screw stirring [23], and the chemical grain refining method [24] are usually used to increase the nucleation rate, promote melt convection and inhibit dendrite growth. Among them, magnetic stirring has attracted extensive attention because of its convenient maintenance, high efficiency, and non-pollution quality. Permanent magnet stirring, one of the magnet stirring methods, has the advantages of long service life, low equipment cost, simple structure, easy maintenance, low power consumption, and recyclable magnetization compared with electromagnetic stirring, which can significantly reduce enterprise costs and improve economic benefits [25]. Yan et al. [26] studied the effect of rotation on forced flow, solidification processes, and microstructures of Sn-3.5% Pb melt in a hollow billet. The results showed that the grain size was refined from 186 to 65 μm when the voltage intensity was 100 V, and it was stated that the rotating magnetic field could reduce the depth of the liquid cavity and improve the microstructure [11]. Zeng et al. [27] investigated the effect of a rotating magnetic field on the solidification of an Sn–Pb alloy and found that increasing the rotating speed and central magnetic field density could effectively reduce surface porosity, refine grain size, and improve tensile properties.

The above evidence shows that permanent magnet stirring has an effect on the microstructures and properties of the alloy. However, the effect of permanent magnet stirring on the casting of Ag-based alloys has rarely been reported. Therefore, in this study, we investigated the effect of permanent magnet stirring on Ag-based alloys using the Ag–1.5Cu–1.0Y alloy. The microstructures, density, microhardness, electrical resistivity, and sulfuration corrosion resistance of the Ag–1.5Cu–1.0Y alloy were tested. The results showed that the permanent magnet stirring refined the dendrites, improved density, microhardness, and sulfuration corrosion resistance of the Ag–1.5Cu–1.0Y alloy. This study provides an experimental basis not only for the improvement of future alloy casting optimization and properties but also for the expansion of the applications of silver-based alloys.

## 2. Materials and Methods

### 2.1. Alloy Design and Preparation

The Ag–1.5Cu–1.0Y alloys used in this study were prepared from high-purity 99.99 wt% Ag, 99.99 wt% Cu, and 99.95 wt% Y (Table 1).

These alloys were prepared by a self-designed open high-frequency induction furnace equipped with an argon (99.99%) atmosphere protection device at 0.1 MP and with permanent magnet stirring. Compared with traditional magnetic stirring, the casting equipment is a high-frequency induction furnace with not only permanent magnetic stirring, but also atmosphere protection, hot top protection, and directional forced condensation (Figure 1a). A graphite crucible was used for the preparation container of the alloys. The permanent magnet was a pair of NdFeB (Figure 1b) with a magnetic field intensity of 2000 GS, and the stirring rate was 0, 300, 600, 900, and 1200 r/min, respectively.

Before melting, the graphite crucible was placed in the melting zone of equipment with the alloying elements loaded. Then, high purity argon was injected into the system for 10 min to ensure that the system was protected by inert gas. Subsequently, the high-frequency electromagnetic induction furnace was opened, and the temperature was slowly raised to 1773 K (1500 °C). The loaded alloy elements were alloyed in a crucible for 10 min when the temperature was stable. Then, the crucible was sent to the casting area and the permanent magnet stirring equipment started with different permanent magnet stirring rates. Each alloy was taken out of the electromagnetic induction furnace after complete cooling and the inert gas environment was maintained throughout the period, so as to prevent the alloy from gas absorption and oxidation during melting and solidification. Finally, the samples were cut, polished, and corroded, and the microstructures and properties of the samples were tested and analyzed using a metallographic microscope, scanning electron microscope (SEM), X-ray electron probe microanalyzer, hardness tester, micro-ohmmeter, and electrochemical workstation.

### 2.2. Microstructures

For microstructure observation, the alloy samples were wet ground with silicon carbide abrasive papers from P800 to P5000, followed by polishing with a solid-to-liquid ratio of 1:4 MgO suspension solution. Then, the polished alloys were corroded in the metallographic corrosive solution (30% H_2_O_2_: 25% NH_3_·H_2_O = 1:3) for 50–60 s to analyze the constituents of the alloys. Microstructure observation was conducted with a Leica optical microscope (Leica, Wetzlar, Germany). In order to illustrate the refining effect of different stirring rates on the dendrite structure of the alloys, the average value of the secondary dendrite arm spacing was generated from the 10 times measurement. Figure 2 shows the schematic diagram of secondary dendrite arm spacing (λ_2_). λ_2_ can be expressed by the average distance of multiple secondary dendrite arm spacing (*L*), as follows [28]:(1)$$\lambda_{2}=\frac{L}{n-1}$$
where λ_2_ is the secondary dendrite arm spacing; *L* is the center-to-center distance of secondary ramifications, and n is the adopted number of ramifications.

### 2.3. X-ray Diffraction (XRD) Phase Constituent Analysis

An X′Pert Pro X-ray diffractometer (Panaco, Eindhoven, The Netherlands) was used to investigate the phase constituents of the Ag–1.5Cu–1.0Y alloys at different rotation speeds. The test temperature was room temperature. The voltage was 40 kV. A Cu target was used. The scanning mode was continuous scanning, and the scanning angle (2*θ*) was 30–120°, the scanning step size was 0.02°, and the scanning time of each step was 5 s.

### 2.4. SEM-EDS Analysis

The surface morphology observation, semi-quantitative analysis of the chemical composition, and element distribution of the alloy samples were conducted using a JXA-8530F field emission electron probe ((JEOL, Tokyo, Japan)) under electron acceleration of 15 keV.

### 2.5. Density Test

The density of the alloy samples was tested using the Archimedes drainage method. Firstly, the weight *W*_1_ of the dry sample in the air was weighed on the precision balance, and then the weight *W*_2_ of the sample in distilled water was weighed. The formula for calculating the density of the sample is as follows:(2)$$d=\frac{\rho W_{1}}{W_{1}-W_{2}}$$
where *W*_1_ is the mass of a dry sample in air, *W*_2_ is the mass of a sample in distilled water, and *ρ* is the density of distilled water.

### 2.6. Microhardness Test

Microhardness tests were performed on an HXS-1000AK microhardness tester (Jingda, Xi’an, China) with a load of 25 g and a holding time of 10 s at room temperature. In order to reduce the test error, five points were selected from the edge to the center of each sample for testing, and the final result was taken as the average value of the test.

### 2.7. Electrical Resistivity Test

Before the electrical resistivity test, the alloy samples were cut into wafers, with a diameter of 9.5 mm and a thickness of 1.5 mm. Electrical resistivity tests were then carried out on a ZY9858 digital micro-ohmmeter (Zhengyang, Shanghai, China) at room temperature.

### 2.8. Sulfuration Corrosion Resistance Test

Sulfuration corrosion resistance tests were conducted in a closed container with H_2_S and the changes were observed at 0.5, 1, 2, 4, 6, and 8 h.

The polarization test was carried out in sodium sulfide solution at the CHI600E electrochemical workstation (Chenhua, Shanghai, China). The concentration of sodium sulfide solution was 0.1mol·L^−1^. A mercury-mercuric oxide electrode (standard electrode potential on a hydrogen scale was 0.098 V) and a platinum electrode were used as the reference electrode and the counter electrode, respectively. The as-cast samples were used as the working electrode. Prior to the testing, the surface of the alloy sample was ground with P5000 silicon carbide abrasive paper and then rinsed with distilled water. The polarization tests were conducted using polarization potential from −0.9 V to −0.6 V at a scan rate of 5 mV·s^−1^.

## 3. Results and Discussions

### 3.1. Microstructures

The dendrites in the Ag–1.5Cu–1.0Y alloy without permanent magnet stirring (at 0 r/min) were thicker, the primary dendrites were well developed, and the secondary dendrite arm spacing was large (Figure 3a). The dendrites of the Ag–1.5Cu–1.0Y alloy were refined gradually with the increase in stirring rate (Figure 3b–e), and the dendrites were best refined in the Ag–1.5Cu–1.0Y alloy with a 900 r/min stirring rate (Figure 3d). However, compared with the alloy at 900 r/min, larger dendrites and secondary dendrite arm spacing (λ_2_) were found in the Ag–1.5Cu–1.0Y alloy with a 1200 r/min stirring rate (Figure 3e). The λ_2_ decreased with the increase in stirring rate (Figure 3f), and it reached the minimum value of 25.12 μm in the Ag–1.5Cu–1.0Y alloy with a 900 r/min stirring rate, then it increased in the Ag–1.5Cu–1.0Y alloy with a 1200 r/min stirring rate (Figure 3f). This was in correspondence with the results of the coarsened dendrite in the Ag–1.5Cu–1.0Y alloy at 1200 r/min of permanent magnet stirring rate, compared with the alloys at 900 r/min.

To further explore the phase constituents and analyze the effect of permanent magnet stirring on the dendrite refinement of the Ag–1.5Cu–1.0Y alloys, the XRD patterns of the as-cast samples with different rotation speeds are presented in Figure 4. As shown in Figure 4, the observed phases are all Ag, and no other phase was observed. This may be due to the low addition of Cu (1.5 wt%) and Y (1.0 wt%) to the Ag-based alloy, which means that the new phase is unable to be detected by XRD. The new phase formed by this small addition of element may need to be further observed by electron microscopy or transmission electron microscopy.

Permanent magnet stirring has a series of benefits compared with the traditional casting process. It makes heat and mass transfer more uniform, primary dendrite growth restrained, secondary dendrite arm spacing shortened, dendrite size refined, and the refining effect becomes more obvious with the increase in stirring rate [29,30].

The refined grains could be explained in the following two ways: first, strong convection occurs in the melt stirred by Lorentz force, resulting in the secondary dendrite root breaking and leading to new crystal core formation, which then contributes to the refined grains; second, the dendrite branching often results in some necking due to segregation, whereas the permanent magnet stirring will lead the branching to break off from the necking. In the subsequent solidification process, residual parts larger that are than critical nuclei will form new crystalline nuclei, which increase the number of effective crystalline nuclei [31,32], thus reducing the grain size. However, the weaker permanent magnet stirring speed effect on the grain refinement in alloys with 1200 r/min compared to alloys with 900 r/min could be explained by the fact that the high speed resulted in the sharp decrease in magnetic flux density, which subsequently weakened the refinement effect of permanent magnet stirring.

### 3.2. Element Distribution

As shown in Table 2, a large amount of Cu and Y elements are enriched in the intergranular region of the Ag–1.5Cu–1.0Y alloy without permanent magnet stirring. This is interpreted as the lack of stirring leading to a fast solidification rate, which resulted in the insufficient diffusion of solute elements and then the enrichment of the added alloy elements in the intergranular. With the increase in stirring rate, the distribution of alloying elements is more uniform, and the segregation rate decreases. Moreover, with the increase in rotation speed, the content of element Y in the crystal increases, and the distribution of element Y is more uniform (Table 2). Especially when the stirring rate is 900 r/min, the Y element content in the crystal is the largest, indicating that the diffusion effect of the element is the best when the stirring rate is 900 r/min. However, an amount of Y enrichment was still observed at the grain boundary no matter how the rotation speed changes, which is may due to the low solid solubility (1.0 wt%) of Y in Ag.

We then conducted a surface scanning analysis of the Ag–1.5Cu–1.0Y alloy sample with a permanent magnet stirring rate of 900 r/min (Figure 5). The results showed that there is micro-segregation in the element Cu (Figure 5c), and the element Y mainly segregates between crystals and only a small amount of Y exists within the crystals (Figure 5d).

### 3.3. Density

As shown in Figure 6, the density of the Ag–1.5Cu–1.0Y alloy samples increased gradually with the increase in stirring rate. When the stirring rate is 900 r/min, the density of the Ag–1.5Cu–1.0Y alloy sample reaches a maximum value of 10.32 g/cm^3^, which is close to the theoretical density value of 10.34 g/cm^3^ (Figure 6). However, the density value does not increase anymore when the stirring rate increases higher than 900 r/min (Figure 6). These results indicated that the density of the Ag–1.5Cu–1.0Y alloy can be improved by the permanent magnet stirring rate. However, the density was not increased with the rotation speed when it arrived at a platform period. This may be explained by the fact that the density is closely related to the compactness of the alloy. Grain refinement improved the strength, altering both the bulk and the surface of alloys [33]. When the rotating speed reaches 900 r/min, the diffusion effect of alloy elements reaches its highest and the dendrite refinement degree is the highest, which leads to the highest density; the diffusion effect and dendrite refinement decrease when the rotating speed reaches 1200 r/min, resulting in the decrease in density of the Ag–1.5Cu–1.0Y alloy.

### 3.4. Microhardness

The microhardness of the alloys increases first and then decreases with the increase in stirring rate (Figure 7a). In detail, the microhardness value of the Ag–1.5Cu–1.0Y alloy without permanent magnet stirring is the smallest (58.41 ± 1.34 HV), and it reaches the maximum value (69.82 ± 1.58 HV) in the alloy with the stirring rate of 900 r/min, then it decreases in the alloy with the stirring rate of 1200 r/min (68.63 ± 1.47 HV) (Figure 7a). The relationship between the microhardness and λ_2_ could be well fitted to the Hall–Petch type relationship [34], which is expressed by the following formula:(3)$$\mathrm{HV}=\mathrm{HV}0_{\;}+k\lambda_{2}^{-0.5}$$
where HV_0_ and *k* are material constants.

As the results show in Figure 7b, the microhardness has a relatively fine correlation with the λ_2_ (R^2^ = 0.878), indicating that the smaller the secondary dendrite arm spacing, the higher the microhardness of the Ag–1.5Cu–1.0Y alloy. As shown, the microhardness and λ_2_ of the alloys follow the Hall–Petch type relationship, except for the alloy without PMS, indicating the microhardness increases with the decrease in λ_2_. The dendrite refinement increased with the permanent magnet stirring rotating speed, and it reached the most refinement at 900 r/min rotating speed. The λ_2_ decreased with rotating speed, and it reached the minimum value of alloys at 900 r/min, then it increased at 1200 r/min, leading to the trend of microhardness increase in alloys under 900 r/min and a decrease in alloys at 1200 r/min.

### 3.5. Electrical Resistivity

The electrical resistivity of the Ag–1.5Cu–1.0Y alloys increases with the increase in stirring rate (Figure 8), leading to a decreased electrical conductivity. However, the overall effect of permanent magnet stirring on the electrical resistance of the alloy sample is not very large, with values ranging from 3.05 × 10^−8^ to 3.39 × 10^−8^ Ω·m. The main reason for the slight increase in electrical resistance may lie in the relationship between electrical resistance and electron transfer. Electrical resistivity is affected by many factors, which can be expressed in the following formula:*ρ* = *ρ*_1_ + *ρ*_2_ + *ρ*_3_ + *ρ*_4_(4)
where *ρ* is total resistance, *ρ*_1_ is the neutron scattering resistance, *ρ*_2_ is dislocation scattering resistance, *ρ*_3_ is interface scattering resistance and *ρ*_4_ is impurity scattering resistance [35,36].

The refined dendrite increased electron scattering [37], and the higher content of Y (the electrical resistivity of Y and Ag were 5.95 × 10^−7^ Ω·m and 1.65 × 10^−8^ Ω·m, respectively) in the grain boundary also resulted in the increase in electrical resistivity of the Ag–1.5Cu–1.0Y alloy. Compared with the alloys at 900 r/min, the refined dendrite decreased and intergranular Y decreased in alloys at 1200 r/min, which means that the electrical resistivity experiences a slight decrease in the Ag–1.5Cu–1.0Y alloy under 1200 r/min stirring speed.

### 3.6. Sulfuration Corrosion Resistance

Sulfuration is a significant factor that could degrade the conductivity and reflectance of Ag-based alloy products [3]. Therefore, in this study, the effect of permanent magnet stirring on sulfuration corrosion resistance of the Ag–1.5Cu–1.0Y alloy was tested.

As shown in Figure 9 and Table 3, the sulfuration corrosion resistance of the Ag–1.5Cu–1.0Y alloys with permanent magnet stirring is better than that of non-permanent magnet stirring alloys. The alloys with a stirring rate of 600 r/min and 900 r/min have better sulfuration corrosion resistance than others. The increased sulfuration corrosion resistance could be explained by the fact that the permanent magnet stirring resulted in the uniform distribution of element Y in alloys under the rotation speeds of 600 r/min and 900 r/min, and that the uniform element Y in the alloys improved the sulfuration corrosion resistance of the Ag–1.5Cu–1.0Y alloy.

The potential dynamic polarization test was carried out in order to better evaluate the effect of different stirring rates on the sulfuration corrosion resistance of the Ag–1.5Cu–1.0Y alloy samples.

Among them, the corrosion current of the Ag–1.5Cu–1.0Y alloy without permanent magnet stirring is the smallest, which is 2.61 × 10^−5^. The corrosion current of the Ag–1.5Cu–1.0Y alloy increased with stirring rate, and it reached the largest in alloys with 900 r/min stirring rate (3.78 × 10^−5^), then decreased in alloys with 1200 r/min (2.66 × 10^−5^) (Figure 10, Table 4). Studies have proved that grain refinement would alter both the bulk and the surface of alloys, including changes in grain boundary density and orientation. These surface changes could affect electrochemical behavior [33]. The better grain refinement of the alloy, the greater the corrosion current [38,39]. Therefore, in this study, the dendrite refinement increased with the stirring rate, leading to the increased corrosion current of the Ag–1.5Cu–1.0Y alloy. The corrosion current reached the largest with the most dendrite refinement. Subsequently, the decreased dendrite refinement of the Ag–1.5Cu–1.0Y alloy results in a decreased corrosion current in alloys with a stirring rate of 1200 r/min.

## 4. Conclusions

The microstructure and property defects of Ag-based alloys limit their applications to a great extent. The performance improvement of their microstructures and properties has become an effective way to expand their applications. In this study, we investigated the effects of permanent magnet stirring on the microstructures and properties of the Ag-alloys based on the Ag–1.5Cu–1.0Y alloy. The results showed that the primary dendrite was shortened and the secondary dendrite arm spacing was reduced to a minimum of 25.12 μm at 900 r/min, indicating that permanent magnet stirring refined the dendrite structure of Ag–1.5Cu–1.0Y alloys. This may be a result of the more uniform distribution of alloy elements by permanent magnet stirring. With regard to the properties, permanent magnet stirring improved the density and microhardness of the Ag–1.5Cu–1.0Y alloys, and the maximum density and microhardness were achieved at 10.32 g/cm^3^ and 69.82 HV, respectively, under the rotation speed of 900 r/min. However, the electrical resistivity of the Ag–1.5Cu–1.0Y alloy increases with the increase in the stirring rate of permanent magnets. It may be caused by the uniform distribution of elements Cu and Y in the matrix, which increased the scattering of electrons and then resulted in the increase in the electrical resistivity of the alloy. The sulfuration corrosion resistance was also enhanced by permanent magnet stirring, with better performance under the rotation speed of 900 r/min. The application of the Ag–1.5Cu–1.0Y alloys prepared by permanent magnet stirring needs further study.

## Figures and Tables

**Figure 1 materials-15-05237-f001:**
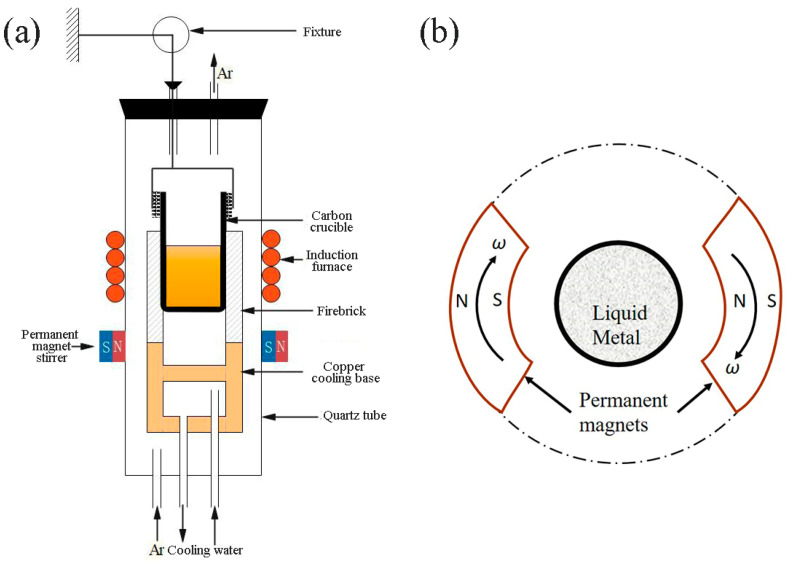
Schematic drawing of the experimental apparatus (**a**). Principle diagram of the magnetic field generated by a pair of NdFeB permanent magnets (**b**).

**Figure 2 materials-15-05237-f002:**
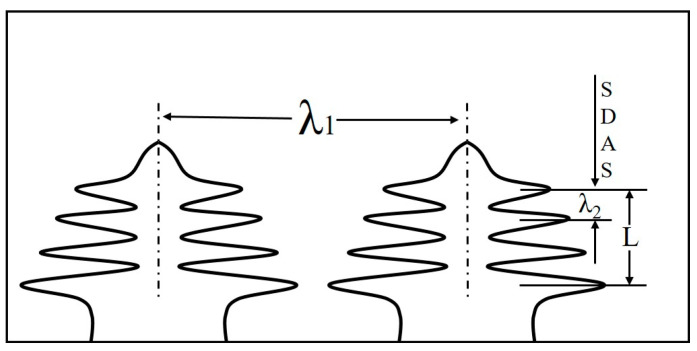
The schematic diagram for measuring the secondary dendrite arm spacing (λ_2_) by section method.

**Figure 3 materials-15-05237-f003:**
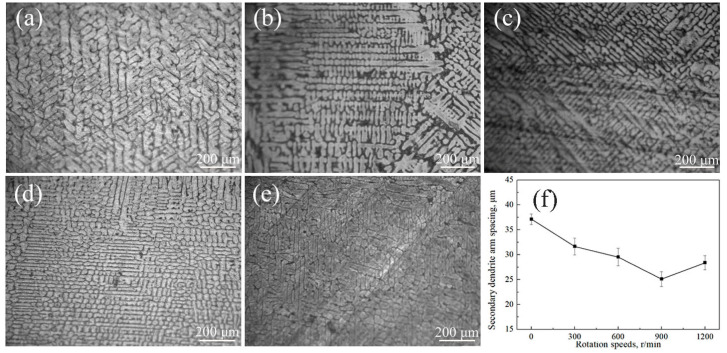
Microstructures of the Ag–1.5Cu–1.0Y alloys at different rotation speeds of 0 r/min (**a**), 300 r/min (**b**), 600 r/min (**c**), 900 r/min (**d**), and 1200 r/min (**e**). Secondary dendrite arm spacing (λ_2_) of the Ag–1.5Cu–1.0Y alloys with different rotation speeds (**f**).

**Figure 4 materials-15-05237-f004:**
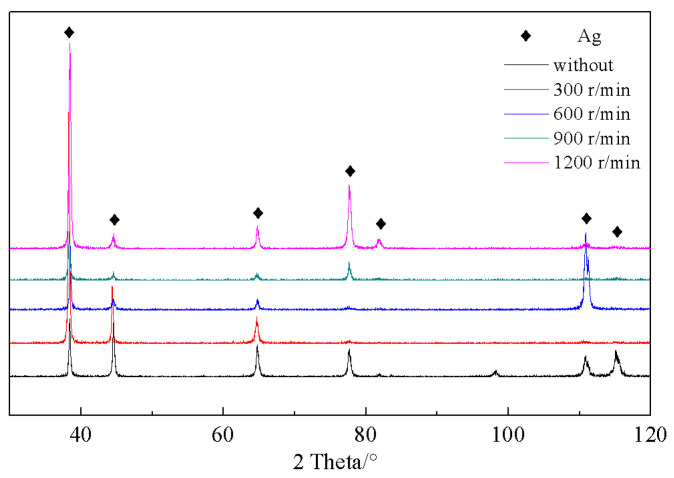
XRD patterns of the Ag–1.5Cu–1.0Y alloys under different rotation speeds.

**Figure 5 materials-15-05237-f005:**
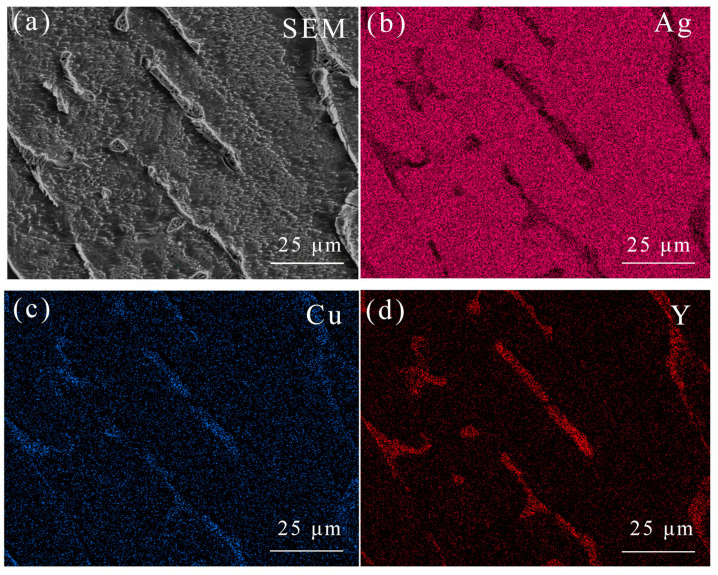
EDS surface scanning of the morphology (**a**), the distribution of Ag (**b**), Cu (**c**), and Y (**d**) in the Ag–1.5Cu–1.0Y alloys under the rotation speed of 900 r/min.

**Figure 6 materials-15-05237-f006:**
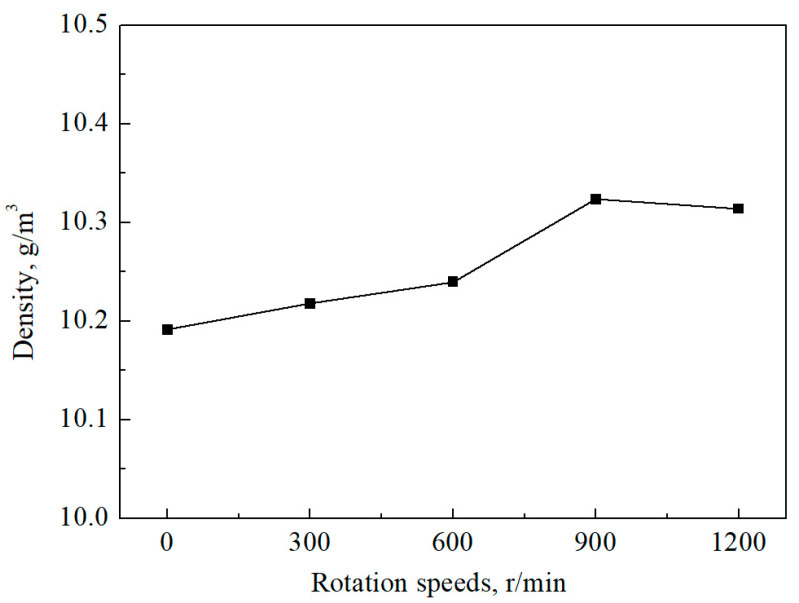
Density of the Ag–1.5Cu–1.0Y alloys obtained under different rotation speeds.

**Figure 7 materials-15-05237-f007:**
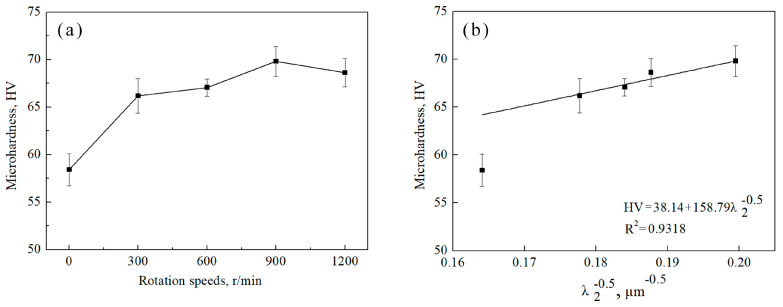
The microhardness of the Ag–1.5Cu–1.0Y alloys under different rotation speeds (**a**) and the linear fitted Hall–Petch relationship between the microhardness and the λ_2_ (**b**).

**Figure 8 materials-15-05237-f008:**
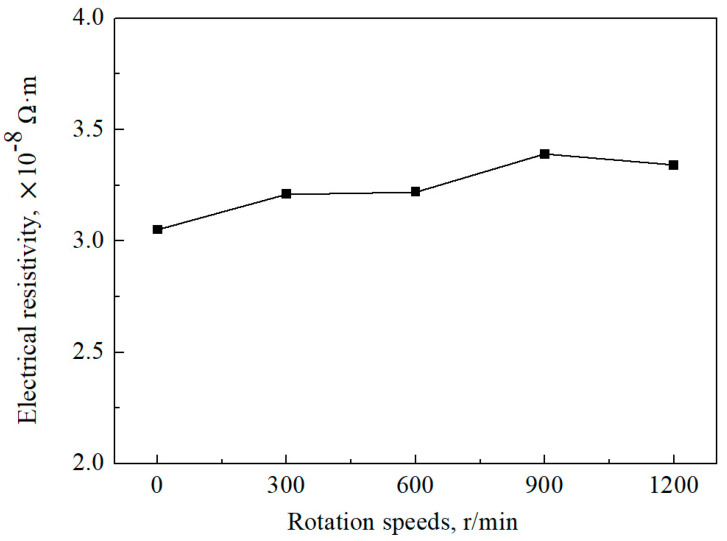
The electrical resistivity of the Ag–1.5Cu–1.0Y alloys under different rotation speeds.

**Figure 9 materials-15-05237-f009:**
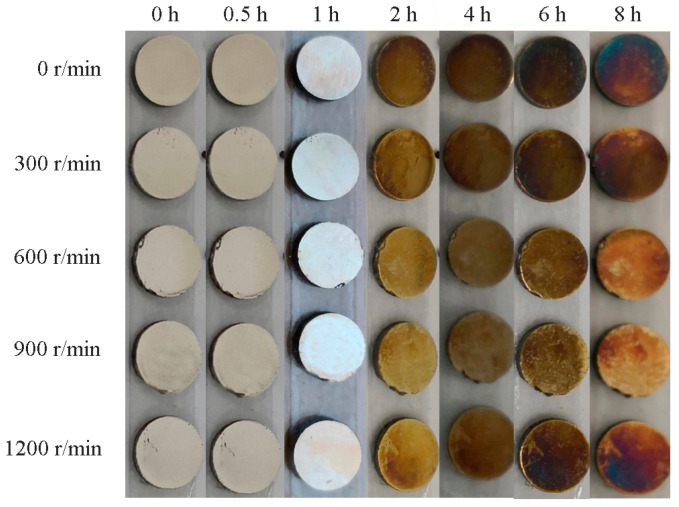
The sulfuration corrosion results of the Ag–1.5Cu–1.0Y alloys under different rotation speeds.

**Figure 10 materials-15-05237-f010:**
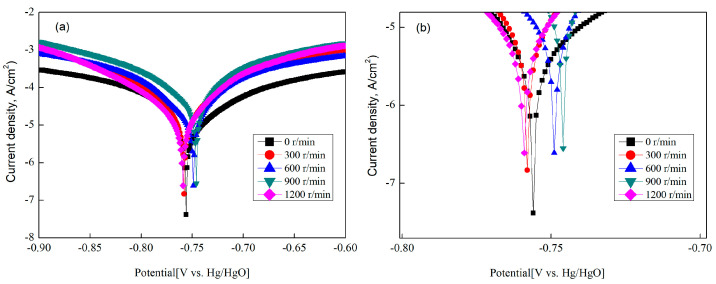
Potential dynamic polarization curves of the Ag–1.5Cu–1.0Y alloys under different rotation speeds (**a**). Zoom in the area between −0.70 and −0.80 V vs. Hg/HgO (**b**).

**Table 1 materials-15-05237-t001:** Chemistry composition (weight percentage, wt%) of the Ag–1.5Cu–1.0Y alloys.

Element	Ag	Cu	Y
wt%	97.5	1.5	1

**Table 2 materials-15-05237-t002:** Element distribution of the Ag–1.5Cu–1.0Y alloys under different rotation speeds (wt%).

Element	0 r/min		300 r/min		600 r/min		900 r/min		1200 r/min	
	a	b	a	b	a	b	a	b	a	b
Ag	97.92	82.95	98.67	91.09	98.89	92.13	98.68	92.31	98.83	90.82
Cu	1.87	5.36	1.19	2.55	0.73	2.27	0.63	2.38	0.70	3.06
Y	0.21	11.69	0.13	6.36	0.38	5.60	0.69	5.31	0.47	6.12

Note: a represents intragranular and b represents intergranular.

**Table 3 materials-15-05237-t003:** Color changes in the sulfuration corrosion in the Ag–1.5Cu–1.0Y alloys under different rotation speeds.

Alloys	0 h	0.5 h	1 h	2 h	4 h	6 h	8 h
0 r/min	Unchanged	Unchanged	Slight yellow	Light yellow	Brown	Purplish brown	Dark brown
300 r/min	Unchanged	Unchanged	Slight yellow	Light yellow	Yellow	Purplish brown	Purplish brown
600 r/min	Unchanged	Unchanged	Slight yellow	Slight yellow	Light yellow	Yellow	Purplish brown
900 r/min	Unchanged	Unchanged	Slight yellow	Slight yellow	Light yellow	Yellow	Purplish brown
1200 r/min	Unchanged	Unchanged	Slight yellow	Light yellow	Yellow	Purplish brown	Purplish brown

Note: corrosion degree: dark brown > purplish brown > brown > yellow > light yellow > slight yellow > unchanged.

**Table 4 materials-15-05237-t004:** Potential dynamic polarization test of the Ag–1.5Cu–1.0Y alloys with different rotation speeds.

Alloys	0 r/min	300 r/min	600 r/min	900 r/min	1200 r/min
*I*corr (A/cm^2^)	2.61 × 10^−5^	3.12 × 10^−5^	3.29 × 10^−5^	3.78 × 10^−5^	2.66 × 10^−5^
*E*corr (V)	−0.756	−0.758	−0.749	−0.746	−0.759

## Data Availability

Not applicable.

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
