# Peer review of "Effect of Magnetic Stirring on Microstructures and Properties of Ag–1.5Cu–1.0Y Alloy"

_materials, 2022, doi:10.3390/ma15155237_

Round 1
Reviewer 1 Report
This work investigates the influence of the permanent magnet stirring method at different rotation speeds on the secondary dendrite arm spacing, dendritic segregation, hardness, and corrosion resistance of Ag-1.5Cu-0.1Y alloys.
The manuscript is well written, and the experimental methods are well presented. For rotation speeds below 1200 rpm, the experimental tests show a direct correlation between the permanent magnet stirring with secondary dendrite arm spacing, dendritic segregation, hardness, and corrosion resistance. However, the trend observed at low rotation speeds modifies at 1200 rpm.
In lines 208 to 212, the authors explain the change in the relationship between the permanent magnet rotation speed and the microstructure. However, this explanation must be extended. In addition, the authors must explain the change in the observed trend for the rest of the properties evaluated.
Author Response
Dear Reviewer,
Thank you very much for all your comments. We would like to submit the revised manuscript entitled "Effect of magnetic stirring on microstructures and properties of Ag-1.5Cu-1.0Y alloy", we wish to be considered for publication in Materials.
The detailed responses are as follows:
The manuscript is well written, and the experimental methods are well presented. For rotation speeds below 1200 rpm, the experimental tests show a direct correlation between the permanent magnet stirring with secondary dendrite arm spacing, dendritic segregation, hardness, and corrosion resistance. However, the trend observed at low rotation speeds modifies at 1200 rpm.
In lines 208 to 212, the authors explain the change in the relationship between the permanent magnet rotation speed and the microstructure. However, this explanation must be extended. In addition, the authors must explain the change in the observed trend for the rest of the properties evaluated.
Response: We appreciate very much for your comments. They are very crucial and helpful in improving the quality of our manuscript. We added the corresponding explanations of the changes in the observed trend for the rest of the properties evaluated according to your suggestions. They mainly include the density on lines 251-257, microhardness on lines 276-283, electrical resistivity on lines 298-303, sulfuration corrosion resistance on lines 312-316, and electrochemical behavior on lines 330-337.
Lines 251-257: This may be explained by the density is closely related to the compactness of the alloy. Grain refinement improved the strength, altering both the bulk and the surface of alloys [33]. When the rotating speed reaches 900 r/min, the diffusion effect of alloy elements reaches its highest and the dendrite refinement degree is the highest, which leads to the highest density; the diffusion effect and dendrite refinement decrease when the rotating speed reaches 1200 r/min, resulting in the decrease in density of the Ag-1.5Cu-1.0Y alloy.
Lines 276-283: As shown, the microhardness and λ2 of the alloys follow the Hall-Petch type relationship except for the alloy without PMS, indicating the microhardness increases with the decrease of λ2. The dendrite refinement increased with the permanent magnet stirring rotating speed, and it reaches the most refinement at 900 r/min rotating speed. The λ2 decreased with rotating speed, and it reached the minimum of alloys at 900 r/min then it increased at 1200 r/min, leading to the trend of microhardness increase in alloys under 900 r/min and a decrease in alloys at 1200 r/min.
Lines 298-303: The refined dendrite increased electron scattering [37], and the higher content of Y (the electrical resistivity of Y and Ag were 5.95×10-7 Ω·m and 1.65×10-8 Ω·m, respectively) in the grain boundary also resulted in the increase in electrical resistivity of the Ag-1.5Cu-1.0Y alloy. Compared with the alloys at 900 r/min, the refined dendrite decreased and intergranular Y decreased in alloys at 1200 r/min, which makes the electrical resistivity a slight decrease in the Ag-1.5Cu-1.0Y alloy under 1200 r/min stirring speed.
Lines 312-316: The increased sulfuration corrosion resistance could be explained by the fact that the permanent magnet stirring uniformed the distribution of element Y in alloys under the rotation speeds of 600 r/min and 900 r/min, and that the uniformed element Y in alloys improved the sulfuration corrosion resistance of the Ag-1.5Cu-1.0Y alloy.
Line 330-337: Studies have proved that grain refinement would alter both the bulk and the surface of alloys, including changes in grain boundary density and orientation. These surface changes could affect electrochemical behavior [33]. The better grain refinement of the alloy, the greater the corrosion current [38, 39]. Therefore, in this study, the dendrite refinement increased with the stirring rate, leading to the increased corrosion current of the Ag-1.5Cu-1.0Y alloy. The corrosion current reached the largest with the most dendrite refinement. Subsequently, the decreased dendrite refinement of the Ag-1.5Cu-1.0Y alloy results in a decreased corrosion current in alloys with a stirring rate of 1200 r/min.
We appreciate your consideration of our manuscript, and all you have done for us.
Sincerely yours,
Desheng Zhang, Hongying Yang and Qin Zhang
Reviewer 2 Report
Review of paper no. materials-1821750 titled Effect of permanent magnetic stirring on microstructures and properties of Ag-1.5Cu-1.0Y alloy by D. Zhang et al.
This is an interesting article that studied the effect of magnetic stirring during melting in an induction furnace on the properties of an Ag alloy. The paper is original as the same alloy has not been studied before. It is publishable subject to revision.
1.I recommend removing the word “permanent” from the title. The magnet was permanent, however, the stirring itself had a definite duration.
2.The authors should specify the concrete applications of the chosen Ag alloy. Why were the Cu and Y alloying elements added to the alloy?
3.Line 134: The accelerating voltage should be 15 keV.
4.The authors used a sodium sulfide solution (Na2S) for corrosion testing (line 155). The choice of this medium should be explained. Indicate the concentration of Na2S. Why have you not used the NaCl solution?
5.You had a mercury-mercuric oxide reference electrode (line 157) which is quite rarely used. Calomel or silver chloride electrodes are more common. Indicate the standard potential of your reference electrode on a hydrogen scale.
6.The paper lacks x-ray diffraction patterns of the prepared materials. The phase constituents of the alloys must be clearly indicated in the manuscript.
7.Corrosion resistance was studied by visual inspection. It would help to include images of the coupons in Table 3.
8.The results in Fig. 9 and Table 4 do not correspond to each other. The alloy without magnetic stirring (black curve) seems to have the lowest corrosion current in Fig. 9. The 900 rpm alloy, on the other hand, seems to have the highest corrosion current (the highest corrosion rate). Zoom in the area between -0.70 and -0.80 V vs. Hg/HgO to make the differences between the alloys clearly visible.
9.The name of the reference electrode must be mentioned in the title of the x axis. Use the following: Potential [V vs. Hg/HgO].
10.Have you measured an open circuit potential of the alloys?
Author Response
Dear Reviewer,
Thank you very much for all your comments. We would like to submit the revised manuscript entitled "Effect of magnetic stirring on microstructures and properties of Ag-1.5Cu-1.0Y alloy", we wish to be considered for publication in Materials.
The detailed responses are as follows:
- I recommend removing the word “permanent” from the title. The magnet was permanent, however, the stirring itself had a definite duration.
Response 1: Thank you very much for your suggestion. We deleted the word “permanent” from the title according to your advice in the revised manuscript.
- The authors should specify the concrete applications of the chosen Ag alloy. Why were the Cu and Y alloying elements added to the alloy?
Response 2: We appreciate your suggestions very much. They are helpful for improving the quality of our manuscript. We added the concrete applications of the chosen Ag alloy to read: “For instance, Ag-Yb and Ag-Mg alloys could be fabricated into plasmonic nanostructures and transparent organic light-emitting diodes [6]. Silver–antimony–boron alloy was applied to electrical contacts, it showed high wear resistance, sulfuration corrosion resistance, and microhardness [7]. Silver–tungsten refractory materials are mainly used in industrial and domestic circuit-breaker products due to their good weld and erosion-resistant properties [8].” on lines 34-39 in the revised manuscript.
And we added the explanation of the addition of Cu and Y alloying elements on lines 43-52 according to your advice in the revised manuscript.
Lines 43-52: Pure Ag has the disadvantages of poor hardness, easy oxidation, and sulfuration. Copper (Cu) could increase the hardness [9, 10], tensile strength [10], and oxidation resistance [11] of the alloy matrix. The addition of rare earth elements can significantly improve the mechanical properties of alloys [12-14]. Yttrium (Y) addition was reported to significantly enhance the toughness and isotropy, and significantly decrease the differences between the longitudinal and transverse textures of the offshore engineering steel [15]. In addition, the addition of Y could enhance the oxidation resistance [16], refinement and homogenization of eutectic phases [17, 18], improve tensile strength, hardness and elongation, the creep rupture life and ductility of the alloy [18], and etc.
- Line 134: The accelerating voltage should be 15 keV.
Response 3: We corrected it to “15 keV” according to your advice.
- The authors used a sodium sulfide solution (Na2S) for corrosion testing (line 155). The choice of this medium should be explained. Indicate the concentration of Na2S. Why have you not used the NaCl solution?
Response 4: Since sulfuration is a significant factor that could degrade the conductivity and reflectance of Ag-based alloy products. Therefore, for the corrosion test, we mainly focus on the sulfuration corrosion resistance. Thus, the sodium sulfide solution (Na2S) was chosen to test the effect of permanent magnet stirring on sulfuration corrosion resistance of the Ag-1.5Cu-1.0Y alloy. We added the explanation of the choice of sodium sulfide solution (Na2S) instead of NaCl solution for corrosion testing on lines 306-308 in the revised manuscript. And we changed the headline of “corrosion resistance” to “sulfuration corrosion resistance” in the corresponding content throughout the manuscript.
Lines 306-308: Sulfuration is a significant factor that could degrade the conductivity and reflectance of Ag-based alloy products [3]. Therefore, in this study, the effect of permanent magnet stirring on sulfuration corrosion resistance of the Ag-1.5Cu-1.0Y alloy was tested.
The concentration of sodium sulfide solution was 0.1mol·L-1, which was indicated on lines 167-168 in the revised manuscript.
- You had a mercury-mercuric oxide reference electrode (line 157) which is quite rarely used. Calomel or silver chloride electrodes are more common. Indicate the standard potential of your reference electrode on a hydrogen scale.
Response 5: Thank you very much for your comments. When we first carried out the experiment, we tried to use the calomel electrode. However, the results were not good. The calomel electrode turns black in the strong alkali solution of sodium sulfide. The stability of the calomel electrode is poor during the test process, and the electrode is also very easy to damage. Therefore, we chose the mercury-mercuric oxide reference electrode after consulting the references. And the standard potential of the mercury-mercuric oxide reference electrode on a hydrogen scale was added on lines 168-169 in the revised manuscript.
- The paper lacks x-ray diffraction patterns of the prepared materials. The phase constituents of the alloys must be clearly indicated in the manuscript.
Response 6: Thank you very much for your advice. We added the x-ray diffraction patterns of the Ag-1.5Cu-1.0Y alloys at different rotation speeds as Figure 4 and the description of corresponding results on lines 192-199 in the revised manuscript.
Lines 192-199: To further explore the phase constituents and analyze the effect of permanent magnet stirring on the dendrite refinement of the Ag-1.5Cu-1.0Y alloys, the XRD patterns of as-cast samples with different rotation speeds are presented in Figure 4. As shown in Figure 4, the observed phases are all Ag, and no other phase was observed. This may be due to the low addition of Cu (1.5 wt%) and Y (1.0 wt%) to the Ag-based alloy, which makes the new phase unable to be detected by XRD. The new phase formed by this small addition of element may need to be further observed by an electron microscopy or transmission electron microscopy.
- Corrosion resistance was studied by visual inspection. It would help to include images of the coupons in Table 3.
Response 7: Thank you very much for the suggestion. We added the images of the coupons in Table 3 as Figure 9 in the revised manuscript.
- The results in Fig. 9 and Table 4 do not correspond to each other. The alloy without magnetic stirring (black curve) seems to have the lowest corrosion current in Fig. 9. The 900 rpm alloy, on the other hand, seems to have the highest corrosion current (the highest corrosion rate). Zoom in the area between -0.70 and -0.80 V vs. Hg/HgO to make the differences between the alloys clearly visible.
Response 8: Thank you very much for your suggestion. We refitted the data and made the corrections in the revised manuscript. And we zoomed in the area between -0.70 and -0.80 V vs. Hg/HgO in Figure 10 to make the differences between the alloys clearly visible according to your advice.
- The name of the reference electrode must be mentioned in the title of the x axis. Use the following: Potential [V vs. Hg/HgO].
Response 9: Thank you very much for your advice. We changed the title of the x-axis in Figure 10 to Potential [V vs. Hg/HgO] according to your suggestion.
- Have you measured an open circuit potential of the alloys?
Response 10: Thank you very much for your enlightening question. Considering that open circuit potential is mainly measured in battery applications, we have not measured the open circuit potential in our work. But we will pay attention to the measurement of the open circuit potential in our subsequent work.
We appreciate your consideration of our manuscript, and all you have done for us.
Sincerely yours,
Desheng Zhang, Hongying Yang and Qin Zhang
Reviewer 3 Report
This review paper deals with the properties of Ag-1.5Cu-1.0Y alloy prepared with NdFeB permanent magnets. Its review is very interesting work in the related field for potential readers. However, there is not clear something for publication and I recommend the author should improve the manuscript.
1. In the Introduction, your research’s motivation or originality should be shown in the Introduction.
2. In figure 2, I recommend that the authors would show more detailed fabrication equipment. And what is different from the equipment compared with conventional magnetic stirring?
3. For microstructures of Ag-1.5Cu-1.0Y alloy, authors should show other data such as crystal size with XRD.
4. As increasing RPM, how about the thickness of Ag-1.5Cu-1.0Y alloy? I recommend you would add a cross-section of SEM images.
5. In Conclusions. I recommend you would summarize the conclusions with your originality.
Author Response
Dear Reviewer,
Thank you very much for all your comments. We would like to submit the revised manuscript entitled "Effect of magnetic stirring on microstructures and properties of Ag-1.5Cu-1.0Y alloy", we wish to be considered for publication in Materials.
The detailed responses are as follows:
- In the Introduction, your research’s motivation or originality should be shown in the Introduction.
Response 1: Thank you very much for your suggestions. They are crucial for improving the quality of our manuscript. We rephrased the section Introduction according to your advice. Major changes include adding the concrete applications of the chosen Ag-based alloys (lines 29-39), the reasons for the Cu and Y alloying elements added to the alloy (lines 43-60), the reason for the chosen method of permanent magnet stirring (lines 61-81), and the meaning of our study (Lines 82-87).
Lines 29-39: More notably, Ag has the highest reflectance in the visible region, thermal conductivities, and the lowest resistivity among all metals [2, 3]. Therefore, it is widely used in the field of target materials. The target materials were then coated on the substrate by the magnetron sputtering technology, used for the production of liquid crystal displays, optical recording media, low-radiation glass electrode film or reflective layer, etc. [4, 5]. For instance, Ag-Yb and Ag-Mg alloys could be fabricated into plasmonic nanostructures and transparent organic light-emitting diodes [6]. Silver–antimony–boron alloy was applied to electrical contacts, it showed high wear resistance, sulfuration corrosion resistance, and microhardness [7]. Silver–tungsten refractory materials are mainly used in industrial and domestic circuit-breaker products due to their good weld and erosion-resistant properties [8].
Lines 43-60: Pure Ag has the disadvantages of poor hardness, easy oxidation, and sulfuration. Copper (Cu) could increase the hardness [9, 10], tensile strength [10], and oxidation resistance [11] of the alloy matrix. The addition of rare earth elements can significantly improve the mechanical properties of alloys [12-14]. Yttrium (Y) addition was reported to significantly enhance the toughness and isotropy, and significantly decrease the differences between the longitudinal and transverse textures of the offshore engineering steel [15]. In addition, the addition of Y could enhance the oxidation resistance [16], refinement and homogenization of eutectic phases [17, 18], improve tensile strength, hardness and elongation, the creep rupture life and ductility of the alloy [18], and etc.
Ag-based alloys’ properties directly affect the quality of the coating film [19]. Moreover, the grain refinement and high density of Ag-based alloys help to improve the efficiency of magnetron sputtering and the quality of alloy film. However, a series of defects, including coarse grains, segregation, porosity, and shrinkage, exist in the traditional casting process of Ag-based alloys. These defects affect the Ag-based alloys’ mechanical properties and service life, leading to a disadvantage in the subsequent processing of the material. The addition of Cu and Y may reduce a part of the above defects. Thus, the Ag-1.5Cu-1.0Y alloy was designed and used in this study.
Lines 61-81: In addition, methods include mechanical stirring [20], permanent magnet stirring [21], electromagnetic stirring [22], twin-screw stirring [23], and the chemical grain refining method [24] are usually used to increase the nucleation rate, promote melt convection and inhibit dendrite growth. Among them, magnetic stirring has attracted extensive attention because of its convenient maintenance, high efficiency, and non-pollution. Permanent magnet stirring, one of the magnet stirring methods, has the advantages of long service life, low equipment cost, simple structure, easy maintenance, low power consumption, and recyclable magnetization compared with electromagnetic stirring, which can significantly reduce enterprise costs and improve economic benefits [25]. Yan et al. [26] studied the effect of rotation on forced flow, solidification process, and microstructures of Sn-3.5% Pb melt in a hollow billet. The results showed that the grain size was refined from 186 to 65 μm when the voltage intensity was 100 V, and he stated that the rotating magnetic field could reduce the depth of the liquid cavity and improve the microstructure [11]. Zeng et al. [27] investigated the effect of a rotating magnetic field on the solidification of Sn-Pb alloy and found that increasing the rotating speed and central magnetic field density could effectively reduce surface porosity, refine grain size, and improve tensile properties.
The above evidence shows that permanent magnet stirring has effects on the microstructures and properties of the alloy. However, the effect of permanent magnet stirring on the casting of Ag-based alloys has rarely been reported. Therefore, in this study, we investigated the effect of permanent magnet stirring on Ag-based alloys using the Ag-1.5Cu-1.0Y alloy.
Lines 82-87: The results showed that the permanent magnet stirring refined the dendrite, improved density, microhardness, and sulfuration corrosion resistance of the Ag-1.5Cu-1.0Y alloy. This study provides an experimental basis not only for the improvement of future alloy casting optimization and properties but also for the expansion of the applications of silver-based alloys.
- In figure 2, I recommend that the authors would show more detailed fabrication equipment. And what is different from the equipment compared with conventional magnetic stirring?
Response 2: Thank you very much for the suggestion. We added a schematic drawing of the experimental apparatus as Figure 1a in the revised manuscript. And we added the difference between the equipment and conventional magnetic stirring to read “Compared with traditional magnetic stirring, the casting equipment is a high-frequency induction furnace with not only permanent magnetic stirring, but also atmosphere protection, hot top protection, and directional forced condensation (Figure 1a).” on lines 98-101 in the revised manuscript.
- For microstructures of Ag-1.5Cu-1.0Y alloy, authors should show other data such as crystal size with XRD.
Response 3: Thank you very much for your advice. We added the x-ray diffraction patterns of the Ag-1.5Cu-1.0Y alloys at different rotation speeds as Figure 4 and the description of the corresponding results on lines 192-199 in the revised manuscript.
Lines 192-199: To further explore the phase constituents and analyze the effect of permanent magnet stirring on the dendrite refinement of the Ag-1.5Cu-1.0Y alloys, the XRD patterns of as-cast samples with different rotation speeds are presented in Figure 4. As shown in Figure 4, the observed phases are all Ag, and no other phase was observed. This may be due to the low addition of Cu (1.5 wt%) and Y (1.0 wt%) to the Ag-based alloy, which makes the new phase unable to be detected by XRD. The new phase formed by this small addition of element may need to be further observed by an electron microscopy or transmission electron microscopy.
- As increasing RPM, how about the thickness of Ag-1.5Cu-1.0Y alloy? I recommend you would add a cross-section of SEM images.
Response 4: Thank you very much for your suggestion. In this study, the alloy samples were melted and cast with a graphite crucible. The crucible has an inner diameter of 9.5 mm, an outer diameter of 17 mm, and a height of 60 mm. After casting, the sample is a cylinder about 20 mm high. In order to facilitate subsequent testing, the original sheet with a diameter of 9.5 mm and a thickness of 1.5 mm is cut. The microstructures were observed by SEM using a cross-sectional thickness of 1.5 mm. The result is shown in Figure 5.
- In Conclusions. I recommend you would summarize the conclusions with your originality.
Response 5: We appreciate your suggestion very much. It is very important to improve the quality of our manuscript. We rewrote the section of Conclusions according to your suggestion on lines 343-361 in the revised manuscript.
Lines 343-361: The microstructure and property defects of Ag-based alloys limit their applications to a great extent. The performance improvement of their microstructures and properties has become an effective way to expand their applications. In this study, we investigated the effects of permanent magnet stirring on the microstructures and properties of the Ag-alloys based on the Ag-1.5Cu-1.0Y alloy. The results showed that the primary dendrite was shortened and the secondary dendrite arm spacing was reduced to a minimum of 25.12 μm at 900 r/min, indicating that permanent magnet stirring refined the dendrite structure of Ag-1.5Cu-1.0Y alloys. That may be a result of the more uniform distribution of alloy elements by permanent magnet stirring. For the properties, permanent magnet stirring improved the density and microhardness of the Ag-1.5Cu-1.0Y alloys, and the maximum density and microhardness were achieved at 10.32 g/cm3 and 69.82 HV, respectively, under the rotation speed of 900 r/min. However, the electrical resistivity of the Ag-1.5Cu-1.0Y alloy increases with the increase in the stirring rate of permanent magnets. It may be caused by the uniform distribution of elements Cu and Y in the matrix, which increased the scattering of electrons and then resulted in the increase in the electrical resistivity of the alloy. The sulfuration corrosion resistance was also enhanced by permanent magnet stirring, with better performance under the rotation speed of 900 r/min. The application of the Ag-1.5Cu-1.0Y alloys prepared by permanent magnet stirring needs further study.
We appreciate your consideration of our manuscript, and all you have done for us.
Sincerely yours,
Desheng Zhang, Hongying Yang and Qin Zhang
Round 2
Reviewer 2 Report
Authors answered my comments and provided reasonable explanations. The paper has been considerably improved and can be accepted for publication.
Reviewer 3 Report
Arthurs responced to all comments and questions. Their manuscript was revised well.